# Prediction of Heart Function and Volume Status in End-Stage Kidney Disease Patients through N-Terminal Pro-Brain Natriuretic Peptide [note 1]

**DOI:** 10.3390/medicina58080975

**Published:** 2022-07-22

**Authors:** Kyung Ho Lee, Inki Moon, Young Seung Oh, Byung Chul Yu, Moo Yong Park, Jin Kuk Kim, Soo Jeong Choi

**Affiliations:** 1Division of Nephrology, Department of Internal Medicine, Soonchunhyang University Bucheon Hospital, Soonchunhyang University College of Medicine, 170 Jomaru-ro, Bucheon 14584, Korea; futurelkh@schmc.ac.kr (K.H.L.); 108254@schmc.ac.kr (Y.S.O.); nephroybc@schmc.ac.kr (B.C.Y.); mypark@schmc.ac.kr (M.Y.P.); medkjk@schmc.ac.kr (J.K.K.); 2Division of Cardiology, Department of Internal Medicine, Soonchunhyang University Bucheon Hospital, Soonchunhyang University College of Medicine, 170 Jomaru-ro, Bucheon 14584, Korea; rokstone@schmc.ac.kr

**Keywords:** chronic kidney failure, renal dialysis, pro-brain natriuretic peptide, heart failure

## Abstract

*Background and Objectives*: N-terminal pro-brain natriuretic peptide (NT-proBNP) is a biomarker used to predict heart failure and evaluate volume status in hemodialysis (HD) patients. However, it is difficult to determine the cutoff value for NT-proBNP in HD patients. In this study, we analyzed whether NT-proBNP helps predict heart function and volume status in HD patients. *Materials and Methods*: This prospective observational study enrolled 96 end-stage kidney disease patients with HD. All patients underwent echocardiography and bioimpedance spectroscopy (BIS) after an HD session. Overhydration (OH) was measured by BIS. Laboratory data were obtained preHD, while serum NT-proBNP was measured after HD. Interventions for blood pressure control and dry weight control were performed, and NT-proBNP was re-assessed after a month. *Results*: There was an inverse correlation between NT-proBNP and ejection fraction (EF) (β = −0.34, *p* = 0.001). OH (β = 0.331, *p* = 0.001) and diastolic dysfunction (β = 0.226, *p* = 0.027) were associated with elevated NT-proBNP. In a subgroup analysis of diastolic dysfunction grade, NT-proBNP increased according to dysfunction grade (normal, 4177 pg/mL [2637–10,391]; grade 1, 9736 pg/mL [5471–21,110]; and grades 2–3, 26,237 pg/mL [16,975–49,465]). NT-proBNP showed a tendency toward a decrease in the ‘reduced dry weight’ group and toward an increase in the ‘increased dry weight’ group compared to the control group (ΔNT-proBNP, −210 pg/mL [−12,899 to 3142], *p* = 0.104; 1575 pg/mL [−113 to 6439], *p* = 0.118). *Conclusions*: We confirmed that NT-proBNP is associated with volume status as well as heart function in HD patients.

## 1. Introduction

More than 50% of end-stage kidney disease (ESKD) patients die from cardiovascular disease. This mortality rate is 10 to 20 times higher than the general population [1]. In one study, the incidence of heart failure (HF) in 1900 hemodialysis (HD) patients was 71 per 1000 person-years, with an 83% mortality rate at 3 years [2].

N-terminal pro-brain natriuretic peptide (NT-proBNP) is associated with cardiovascular morbidity in patients with HF [3,4]. NT-proBNP lower than the cutoff value is a negative predictor of HF [5,6]. Several studies have also shown that NT-proBNP may help predict left ventricular hypertrophy (LVH) and systolic dysfunction in ESKD patients [7,8]. Because volume overload in ESKD patients can lead to systemic hypertension and HF [9], it is important to properly evaluate volume status. NT-proBNP could be helpful for monitoring excess fluid volume and dry weight [10,11,12]. However, kidney function has an inverse correlation with NT-proBNP level [13,14]. NT-proBNP in ESKD patients is much higher than the normal cutoff values and varies according to the degree of residual renal function [15]. Because NT-proBNP decreases after dialysis compared to before dialysis [3], it is difficult to interpret NT-proBNP values in HD patients. One study demonstrated an association between NT-proBNP and volume overload in patients with hemodialysis but not cardiac dysfunction [10]. In another study [11], although NT-proBNP showed a correlation with both volume status and cardiac function, there was a limitation in that volume overload was overestimated by measuring bioimpedance spectroscopy and NT-proBNP on a midweek non-dialysis day.

Unlike previous studies, we measured NT-proBNP at the end of HD (midweek) to evaluate the clinical usefulness of NT-proBNP in more consistent volume status and less overhydration. We divided overhydration, EF, and diastolic dysfunction into subgroups and analyzed association with NT-proBNP. This study was conducted under the hypothesis that NT-proBNP level would be related to both heart function and volume status in HD patients.

## 2. Materials and Methods

### 2.1. Patients

From June to August 2021, we conducted a prospective observational study of HD patients. Patient recruitment took place at the Soonchunhyang University Bucheon Hospital, Korea. Patients over 20 years of age and 6 months or more after starting HD were included. Patients with pre-existing heart disease were also included. Patients receiving concurrent peritoneal dialysis, patients with a life expectancy of less than 3 months, and patients who refused echocardiography, bioimpedance spectroscopy (BIS), or NT-proBNP testing were excluded. Patients who have undergone renal transplantation during follow-up were also excluded.

All patients underwent a physical examination, electrocardiogram, transthoracic echocardiography (TTE), and BIS. Patients were dialyzed with the same dialysis machine and received regular bicarbonate HD three times a week in sessions lasting 3.5 to 4 h. Dialysis adequacy was calculated for every patient on a monthly basis using Kt/V (K = dialyzer clearance, t = time, V = volume of water a patient’s body contains), using the Daugirdas formula with the single pool method of blood urea measurement [16].

Ischemic heart disease is defined as an acute coronary syndrome with prior percutaneous coronary intervention or anti-ischemic medication. HF was defined by the guidelines of the European Society of Cardiology as symptoms (dyspnea on exertion or ankle swelling) and echocardiography findings suggesting cardiac dysfunction [17].

All study participants provided written informed consent. The study was approved by the ethics committee (No. 2021-07-034-001) and was performed according to the principles expressed in the Declaration of Helsinki.

### 2.2. Laboratory Measures

Baseline demographic and medical histories were recorded. Blood samples were taken before dialysis on the day of a mid-week dialysis session. The patient’s vital signs were measured in a stable state and at the start of dialysis. Only serum NT-proBNP was taken at the end of HD according to the dry weight of patients before discontinuation of the extra-corporeal circulation. One month later, NT-proBNP was re-assessed in the same manner.

NT-proBNP was measured using Elecsys proBNP II (Roche Diagnostics, Mannheim, Germany). The cutoff used was 125 pg/mL based on the manufacturer’s instructions. The detection range was 5 to 100,000 pg/mL and the coefficient of variation (20%) was <50 pg/mL. Blood samples were stored at 2 to 8 °C.

### 2.3. Measurement of Echocardiographic Parameters

All echocardiographic studies were performed by an experienced clinical sonographer using commercially available ultrasound equipment (E90/95; GE Healthcare, Milwaukee, WI, USA) on a non-dialysis date. Two-dimensional echocardiography, continuous- and pulsed-wave Doppler measurements, and tissue Doppler imaging were obtained using standard techniques and procedures according to the American Society of Echocardiography guidelines [18].

Left ventricular mass (LVM) was calculated according to the regression equation described by Devereux: LVM = 0.832 (IVSD + LVEDD + PWTD)^3^ − (LVEDD)^3^ + 0.6 and indexed to height^2.7^ to determine LVM index (LVMI), presented in g/m^2.7^. LVH was defined as LVMI > 50 g/m^2.7^ in male subjects and > 47 g/m^2.7^ in female subjects.

Body composition monitor (Fresenius Medical Care, Bad Homburg, Germany) measurements based on BIS were used to calculate overhydration (OH). BIS was performed after dialysis (in the middle of week) and after at least 2 min in the supine position to allow sufficient time for re-distribution of fluids.

### 2.4. Definition of Diastolic Dysfunction

Diastolic dysfunction was classified based on the 2016 American Society of Echocardiography guidelines [19]. At first, diastolic dysfunction was classified as present, absent, or intermediate, based on ejection fraction (EF) and four other parameters (E/e’ > 14, lateral e’ < 10 cm/s, tricuspid regurgitant jet velocity > 2.8 m/s, and left atrium volume index > 34 mL/m^2^). Diastolic dysfunction was considered present if more than two of the other four parameters were positive. Patients with EF ≥ 50% and equivocal other parameters were categorized as indeterminate, and diastolic function was considered normal if ≥ two of the four other parameters were negative. Finally, patients with diastolic dysfunction were categorized into severity grades 1 to 3. The severity of diastolic dysfunction was graded as follows: E/A ratio ≤ 0.8 and E ≤ 50 cm/s, grade 1 dysfunction; E/A > 2, grade 3 dysfunction. Intermediate cases were classified according to whether they had E/e’ > 14, tricuspid regurgitant jet velocity > 2.8 m/s, and left atrium volume index > 34 mL/m^2^. If two of these features were positive, patients were classified as having grade 2 diastolic dysfunction. If two were negative, they were classified as having grade 1 diastolic dysfunction.

### 2.5. Statistical Analysis

All data with a normal distribution are expressed as mean with standard deviation, otherwise as median with the range or interquartile range in parenthesis. A Mann–Whitney U test was used for comparison of NT-proBNP according to categorical data. Multiple linear regression analysis was used to assess the combined influence of variables on serum NT-proBNP concentration. The association between NT-proBNP and heart function was analyzed by subgroups according to NT-proBNP level using Fisher’s exact test. The Kruskal–Wallis test and Mann–Whitney U test were used to compare ΔNT-proBNP (1 month after NT-proBNP–initial NT-proBNP) according to the intervention group. A value of *p* less than 0.05 was considered statistically significant. All statistical analyses were performed using SPSS version 11.5 (SPSS Inc., Chicago, IL, USA).

## 3. Results

### 3.1. Characteristics of the Study Patients

Table 1 shows the baseline characteristics of 96 study participants undergoing HD. The median duration of HD treatment was 86 months (4–352). In the total population, 91.7% and 42.7% had a history of hypertension and diabetes, respectively. A total of 26% of participants had a history of congestive HF (CHF). Median serum NT-proBNP of participants measured post-HD was 11,576 pg/mL [4058–22,460]. Median OH measured by BIS was 0.4 L [−0.7 to 1.5]. On TTE, EF was 59.2 ± 8.2%; 42.7% of participants showed LVH; 63.2% of the participants had diastolic dysfunction.

### 3.2. Comparison of NT-proBNP Levels According to Variables

Table 2 compares categorical variables and NT-proBNP levels. There was no significant difference in NT-proBNP according to sex or diabetes history. Patients with a history of CHF had significantly higher NT-proBNP (18,743 pg/mL [12,385–50,154] vs. 8931 pg/mL [3170–17,413], *p* < 0.001). Patients with atrial fibrillation and wall motion abnormalities also showed significant differences in NT-proBNP compared to those without. NT-proBNP was significantly higher in the group with LVH compared to the group without 21,110 pg/mL [12,665–40,790] vs. 6006 pg/mL [2637–11,955], *p* < 0.001). The group with diastolic dysfunction also had significantly higher NT-proBNP (17,129 pg/mL [7771–29,075] vs. 4177 pg/mL [2637–10,391], *p* < 0.001).

### 3.3. Risk Factors for Higher NT-proBNP Level

Multivariate linear regression analysis showed a significant inverse correlation between NT-proBNP and EF (β = −0.34, *p* = 0.001) (Table 3). BIS-OH (β = 0.331, *p* = 0.001) and the presence of diastolic dysfunction (β = 0.226, *p* = 0.027) were associated with elevated NT-proBNP. Duration of dialysis treatment did not show a significant correlation with NT-proBNP, while hemoglobin showed a positive correlation tendency only in univariate analysis (β = −0.183, *p* = 0.074).

In Figure 1, three variables (diastolic dysfunction, BIS-OH, and EF) showed a significant association with NT-proBNP when divided into subgroups. Compared to normal diastolic function, NT-proBNP level increased rapidly from grade 1 to grades 2 and 3 of diastolic dysfunction (26,237 pg/mL [16,975–49,465]). Tertiles of OH about NT-proBNP levels are shown. NT-proBNP level increased significantly with OH ≥ 1 L (18,743 pg/mL [9626–40,790]). As EF decreased, NT-proBNP level increased and was significantly increased when EF was 55% or less (23,333 pg/mL [8782–71,114]).

### 3.4. Cutoff Level of NT-proBNP for HF

In Table 4, the association between heart function on TTE and NT-proBNP was analyzed by dividing participants based on 4058 pg/mL and 11,576 pg/mL corresponding to the 25th percentile and the 50th percentile (median) of NT-proBNP of all participants. When NT-proBNP was higher than 4058 pg/mL, it was significantly associated with diastolic dysfunction and the presence of LVH. In addition, when NT-proBNP was higher than 11,576 pg/mL, it was significantly associated with diastolic dysfunction, the presence of LVH, and wall motion abnormalities, and it was associated with EF < 55% (*p* = 0.07).

### 3.5. Change in NT-proBNP after Intervention

NT-proBNP was re-assessed one month later, and there was no significant difference from initial NT-proBNP (*p* = 0.707) (Figure 2). According to BIS-OH, TTE findings, and BP, participants were grouped into an intervention group (group 1, reduce dry weight; group 2, add BP medications; group 3, increase dry weight) or a group without intervention (group 4, control group). NT-proBNP changes (ΔNT-proBNP, 1 month after NT-proBNP–initial NT-proBNP) are compared in Table 5. There was a significant difference in ΔNT-proBNP among groups (*p* = 0.016). The ΔNT-proBNP interquartile range of group 4 showed relatively little variability compared to other groups (−1090 to 3585 pg/mL). The ΔNT-proBNP of group 1 showed a tendency toward being lower than group 4 (*p* = 0.104), and the ΔNT-proBNP of group 3 showed a tendency toward being higher than group 4 (*p* = 0.118). Group 2 showed a significant decrease in ΔNT-proBNP compared to group 4 (*p* = 0.007). Of the 11 participants in group 2, 7 took β-blockers and 4 took angiotensin II receptor blockers (ARBs). Participants who took ARBs showed a further decrease in NT-proBNP (ARB, −23,831 pg/mL vs. β-blockers, 515 pg/mL; median values, *p* = 0.014) (Appendix A).

## 4. Discussion

Formed by cleavage of proBNP and biologically inactive amino-acid peptide, NT-proBNP is a cardiac biomarker secreted by ventricular myocyte stretching [20]. Because NT-proBNP has a longer half-life compared to BNP, it is more stable [21]. However, in HD patients, NT-proBNP levels decrease after HD [3] and may be affected by the patient’s residual renal function [15]. In addition, BP, volume status, and hemoglobin in HD patients frequently change, which may make NT-proBNP variable.

Unlike TTE or BIS, NT-proBNP testing is inexpensive and easy to perform with a blood sample during dialysis. Serial trends can be evaluated because it is possible to measure repeatedly within a short time interval. It can also be helpful in evaluating both heart function and volume status in patients with HD. In this study, NT-proBNP was sampled post-dialysis for follow-up in the constant volume status (dry weight) as possible. Moreover, we planned to analyze the correlation by measuring NT-proBNP and BIS-OH in dry weight (less overhydration status) after HD. In the previous study [11], because BIS and NT-proBNP were measured on a midweek non-dialysis day, the volume status of the patients was overestimated compared to our study, and the variability of OH was large (OH: 2.68 [−1.9–12.2] vs. 0.4 [−0.7–1.5]). Although NT-proBNP decreased after dialysis, enrolled patients used the same dialyzer and dialysis machine.

Duration of dialysis treatment was expected to show an inverse correlation with NT-proBNP because it is related to residual urine function. However, that was not the case. This is probably because the duration of dialysis treatment of all participants was 86 months (4–352), and most patients had minimal residual urine function. As in the previous meta-analysis study, participants in our study had a longer dialysis duration than other studies. It is significant that NT-proBNP was associated with both heart function and volume status in this patient group.

HF with preserved EF is common; it occurs in more than 50% of HD patients [22]. Diastolic dysfunction occurs in HF with preserved EF and is diagnosed as a TTE finding [19]. NT-proBNP increased as diastolic dysfunction grade increased, and diastolic dysfunction showed a correlation at levels of NT-proBNP ≥ 4058 pg/mL. Although some studies showed a relationship between diastolic dysfunction and NT-proBNP [23,24], to the best of our knowledge, this is the first study on the association between dysfunction grade and NT-proBNP in HD patients.

When EF was divided into subgroups, NT-proBNP level increased as EF decreased, and NT-proBNP level increased significantly when EF was 55% or less. When NT-proBNP was more than 11,576 pg/mL, it was associated with EF < 55% (*p* = 0.07) and was significantly associated with diastolic dysfunction, LVH, and wall motion abnormalities. There have been some studies [7,8] demonstrating the association between EF and NT-proBNP in ESKD patients. Madsen et al. [3] reported that when EF was divided into subgroups in ESKD patients, NT-proBNP increased as EF decreased. In this study, NT-proBNP was 7733 pg/mL in patients with an EF < 44%. However, in our study, median NT-proBNP was 23,333 pg/mL in patients with an EF < 55%. This difference may be influenced by other variables, but the duration of dialysis treatment of the patients included in our study was longer than in previous studies (86 months (4–352) vs. 20 months [1–216]).

As described above, it is very difficult to evaluate volume status in dialysis patients because they are influenced by multiple factors. Previous studies did not demonstrate a significant association between NT-proBNP and volume overload (inter-dialytic weight gain) [3]. One study found no association between extracellular water (%) and NT-proBNP in patients receiving peritoneal dialysis [25]. In case of pre-dialysis CKD patients, many studies have shown that OH measured by BIS and NT-proBNP were correlated [26,27,28]. In our study, when OH was analyzed by subgroup, there was a significant association with NT-proBNP in HD patients. In particular, when considering other studies [29,30,31] that set the fluid overload standard to 1.1 L or higher, the result of a large increase in NT-proBNP at OH ≥ 1 L was significant.

After one month of follow-up, NT-proBNP showed no significant difference from initial NT-proBNP, suggesting the stability and reproducibility of NT-proBNP. In particular, even when the groups were divided according to intervention, as shown in Table 5, the ΔNT-proBNP (1 month after NT-proBNP–initial NT-proBNP) interquartile group 4 (without intervention) showed relatively little variability compared to the other intervention groups.

Although the degree of increase or decrease of dry weight was not constant and is not statistically significant, when the patient’s dry weight was decreased, NT-proBNP tended to decrease compared to group 4, and when dry weight was increased, NT-proBNP tended to increase. It is thought that NT-proBNP reflects patient volume status. Although the number of patients was small (*n* = 11), the group placed on BP medications also showed a significant decrease in NT-proBNP, and in particular, the participants taking ARBs showed a further decrease in NT-proBNP (vs. β-blockers). A study [32] showing that NT-proBNP was higher in patients with high BP suggested that BP control may also help reduce NT-proBNP.

In this study, patients with various dialysis periods were included, and NT-proBNP was sampled post-dialysis for follow-up at as constant a volume status (dry weight) as possible to improve accuracy. There have been no previous studies showing the association between NT-proBNP and subgroups of OH measured by BIS in HD patients (ESKD patients). We divided the diastolic dysfunction grade and confirmed the association with NT-proBNP. The dialysis duration of the enrolled patients was longer. One sonographer performed the TTE to eliminate bias due to sonographer diversity. Moreover, we showed the association with NT-proBNP by a subgroup of diastolic dysfunction.

The limitations of this study were that the total number of participants was small, and not all variables could be controlled because NT-proBNP was multifactorial. The follow-up period of NT-proBNP was short and the degree of increase or decrease of dry weight was not constant. Moreover, the number of patients who took BP medications was small and the type or dose of medications was not consistent. In addition, since follow-up TTE was not performed, whether the observed decrease in NT-proBNP indicated improvement in HF remains unclear.

## 5. Conclusions

We confirmed that NT-proBNP was associated with OH in BIS as well as heart function values such as diastolic dysfunction and EF. As such, NT-proBNP is a biomarker helpful for predicting heart function and volume status in ESKD patients. In the future, a large prospective study with a longer follow-up period is needed.

## Figures and Tables

**Figure 1 medicina-58-00975-f001:**
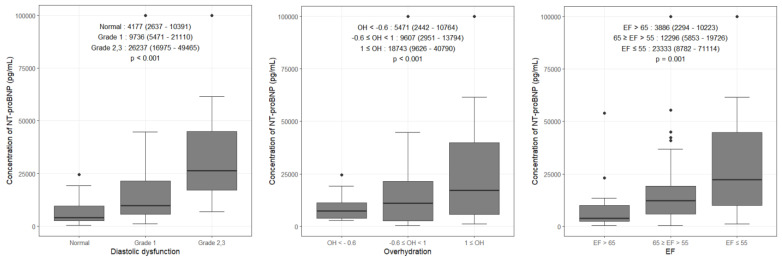
Serum concentrations of N-terminal pro-brain natriuretic peptide (NT-proBNP) graphically shown for variables (diastolic dysfunction, overhydration, ejection fraction) significantly affecting levels in a multivariate linear regression analysis. Values are expressed as median with interquartile range.

**Figure 2 medicina-58-00975-f002:**
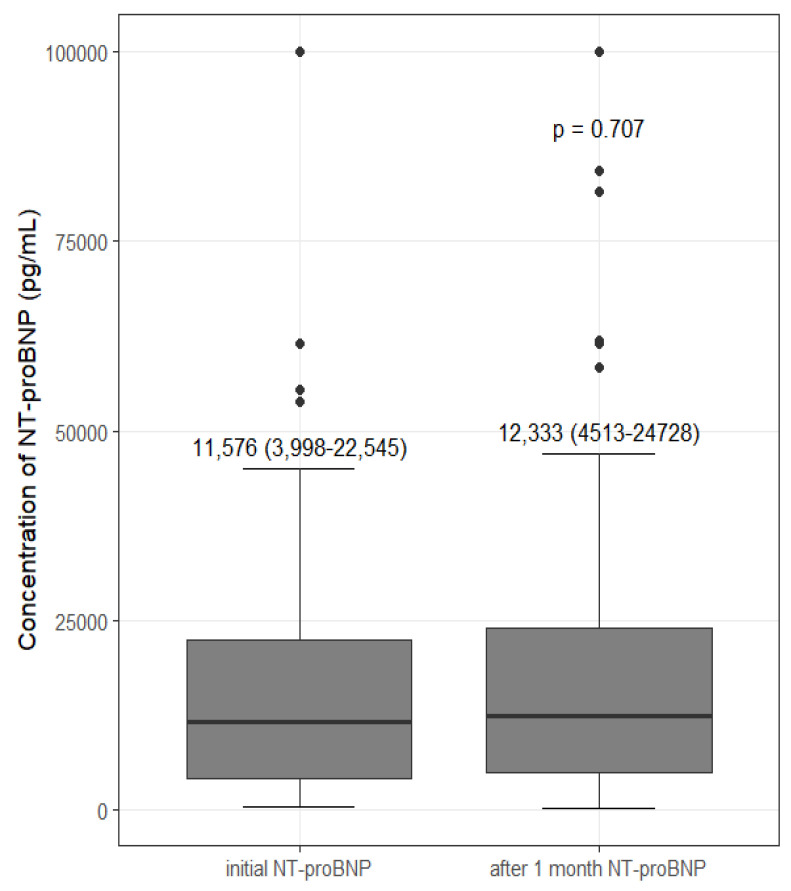
Initial and after 1 month concentrations of N-terminal pro-brain natriuretic peptide (NT-proBNP). Values are expressed as median with interquartile range.

**Table 1 medicina-58-00975-t001:** Baseline characteristics of 96 study participants.

Variable	Values
Age (years)	60.8 ± 11.7
Sex (M/F)	49/47
Body mass index (kg/m^2^)	22.42 ± 3.77
Duration of dialysis treatment (months)	86 (4–352)
History of hypertension (%)	91.7
History of diabetes (%)	42.7
History of IHD (%)	19.8
History of CHF (%)	26
Atrial fibrillation (%)	2.1
Systolic BP (mmHg)	155 ± 18
Diastolic BP (mmHg)	72 ± 13
Heart rate (bpm)	77 ± 9
Hemoglobin (mg/dL)	10.7 ± 0.8
Albumin (g/dL)	4.0 ± 0.4
Phosphate (mg/dL)	4.8 ± 1.0
NT-proBNP (pg/mL)	11,576 (4058–22,460)
Kt/V	1.65 ± 0.2
BIS-OH (L)	0.4 (−0.7–1.5)
EF (%)	59.2 ± 8.2
Wall motion abnormality (%)	12.5
LVH (%)	42.7
Diastolic dysfunction (%)	63.2

Values are expressed as *n*, mean ± standard deviation, median (range), median (interquartile range). M, male; F, female; IHD, ischemic heart disease; CHF, congestive heart failure; BP, blood pressure; NT-proBNP, N-terminal pro-brain natriuretic peptide; Kt/V, K = dialyzer clearance, t = time, V = volume of water a patient’s body contains; BIS-OH, bioimpedance spectroscopy-overhydration; EF, ejection fraction; LVH, left ventricular hypertrophy.

**Table 2 medicina-58-00975-t002:** Categorical variables and N-terminal pro-brain natriuretic peptide level.

Variable	Yes	No	*p*-Value
Male sex	12,261 (3939–22,631)	9736 (3599–22,744)	0.565
History of hypertension	12,274 (4179–23,698)	7724.5 (3886–9348.5)	0.118
History of diabetes	12,261 (2879–24,378)	9736 (4177–18,563)	0.897
History of IHD	17,033 (3017–24,378)	9736 (4058–21,967)	0.448
History of CHF	18,743 (12,385–50,154)	8931 (3170–17,413)	<0.001
Atrial fibrillation	64,716 (29,432–100,000)	10,794.5 (3939–21,304)	0.045
Wall motion abnormality	29,027 (13,550–100,000)	9673 (38,41–18,698)	0.001
LVH	21,110 (12,665–40,790)	6006 (2637–11,955)	<0.001
Diastolic dysfunction	17,129 (7771–29,075)	4177 (2637–10,391)	<0.001

Values are expressed as median (interquartile range). IHD, ischemic heart disease; CHF, congestive heart failure; LVH, left ventricular hypertrophy.

**Table 3 medicina-58-00975-t003:** Factors affecting the N-terminal pro-brain natriuretic peptide in hemodialysis patients.

Independent Variable	*p*-Value	β
Age (years)	0.827	−0.021
Sex	0.007	0.350
Body mass index	0.377	−0.104
History of diabetes	0.695	−0.040
Duration of dialysis treatment	0.265	−0.110
Hemoglobin	0.778	0.027
Kt/V	0.104	−0.234
BIS-OH	0.001	0.331
EF	0.001	−0.340
Diastolic dysfunction	0.027	0.226

Kt/V, K = dialyzer clearance, t = time, V = volume of water a patient’s body contains; BIS-OH, bioimpedance spectroscopy-overhydration; EF, ejection fraction.

**Table 4 medicina-58-00975-t004:** Heart function according to NT-proBNP level.

Variable	NT-ProBNP		*p*-Value	NT-ProBNP		*p*-Value
	<4058 (*n* = 24)	≥4058 (*n* = 72)		<11,576 (*n* = 48)	≥11,576 (*n* = 48)	
Age (years)	0.827	−0.021				
EF (%)			0.174			0.070
<55	1 (4.2)	12 (16.7)		3 (6.3)	10 (20.8)	
≥55	23 (95.8)	60 (83.3)		45 (93.8)	38 (79.2)	
Diastolic dysfunction			<0.001			<0.001
Yes	7 (29.2)	53 (73.6)		20 (41.7)	40 (83.3)	
No	17 (70.8)	18 (25.0)		28 (58.3)	7 (14.6)	
Wall motion abnormality			0.284			0.027
Yes	1 (4.2)	11 (15.3)		2 (4.2)	10 (20.8)	
No	23 (95.8)	61 (84.7)		46 (95.8)	38 (79.2)	
LVH			0.004			<0.001
Yes	4 (16.7)	37 (51.4)		7 (14.6)	34 (70.8)	
No	20 (83.3)	35 (48.6)		41 (85.4)	14 (29.2)	

NT-proBNP, N-terminal pro-brain natriuretic peptide; EF, ejection fraction; LVH, left ventricular hypertrophy.

**Table 5 medicina-58-00975-t005:** Comparison of ΔNT-proBNP according to intervention.

	Total (*n* = 96)	Group 1 (*n* = 17) ^a^	Group 2 (*n* = 11) ^b^	Group 3 (*n* = 23) ^c^	Group 4 (*n* = 45) ^d^
ΔNT-proBNP ^e^	306 (−1341–3662)	−210 (−12,899–3142)	−887 (−6739–2925)	1575 (−113–6439)	330 (−1090–3585)
*p*-Value	0.016	0.104 ^f^	0.007 ^g^	0.118 ^h^	-

Values are expressed as median (interquartile range). NT-proBNP, N-terminal pro-brain natriuretic peptide; ^a^ Group 1: reduce dry weight; ^b^ Group 2: add blood pressure medications; ^c^ Group 3: increase dry weight; ^d^ Group 4: no intervention (control group); ^e^ ΔNT-proBNP = after 1 month NT-proBNP–initial NT-proBNP; ^f^ Group 1 vs. Group 4; ^g^ Group 2 vs. Group 4; ^h^ Group 3 vs. Group 4.

## Data Availability

The data that support the findings of this study are available from the corresponding author (S.J.C.) upon reasonable request.

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
