# Peer review of "Prediction of Heart Function and Volume Status in End-Stage Kidney Disease Patients through N-Terminal Pro-Brain Natriuretic Peptide†"

_medicina, 2022, doi:10.3390/medicina58080975_

Round 1

Reviewer 1 Report

1) “End-stage renal disease” should be replaced with “end-stage kidney disease” in the entire manuscript 

2) The introduction should reflect current knowledge regarding the association between NT-proBNP and volume status and heart failure. There were published more articles on the utility of NT-proBND in end-stage kidney disease, including for prediction of volume status and cardiac function in hemodialysis patients (DOI: https://doi.org/10.1681/ASN.2008010012, doi: 10.1038/sj.ki.5002087, https://www.ncbi.nlm.nih.gov/pmc/articles/PMC6105012/,https://www.ncbi.nlm.nih.gov/pmc/articles/PMC2879314/, https://doi.org/10.3389/fcvm.2021.646402). Thus, the authors should also highlight the novelty of the study.

3) The inclusion and exclusion criteria should be more specific (e.g., it is not clear whether patients with pre-existing heart disease were included or excluded from the study, which could affect the final analysis)

4) The authors should explain why NT-proBNP sample was taken at the end of hemodialysis session (NT-proBNP could be modified after hemodialysis: https://doi.org/10.5301%2Fijao.5000387)

5) In the “Measurement of Clinical Parameters” section the authors reported echocardiographic parameters and no clinical signs. Thus the section title should be modified accordingly.

6) “Diastolic dysfunction was considered present if EF was ≥ 50% and more than two of the other four parameters were positive” – it is not clear why diastolic dysfunction was considered if LVEF was ≥ 50%, as it could be present irrespective of left ventricular systolic function. 

7) Conclusions should be reported as an individual section, and not as a part of discussions.

Author Response

Thank you for review.

Reviewer 2 Report

This is an observational study attempting to correlate NTPro BNP with heart function and volume status post dialysis. The findings confirm that NTProBNP is higher in those with heart failure/LVH/Diastolic dysfunction. I suggest that in introduction and discussion authors should inform about novelty of this finding. These findings are no doubt interesting but practicality of performing this test, at what frequency it needs to be done and whether it can differentiate between heart failure and overhydration may be added. Methodology needs to clarify on what basis patients were allotted to 4 groups including by how much was dry weight reduced and why BP medicines were added to only a portion of participants when 91% had high BP.

Author Response

Thank you for review.

Round 2

Reviewer 1 Report

Great answers. Thank you!

Congrats!